# Multi-modal Variational Encoder-Decoders

**Iulian V. Serban**[†,*] **Alexander G. Ororbia II**[×,*] **Joelle Pineau**[‡]**, Aaron Courville**[†]
[†] Department of Computer Science and Operations Research, Universite de Montreal
[×] College of Information Sciences & Technology, Penn State University
[‡] School of Computer Science, McGill University
iulian[DOT]vlad[DOT]serban[AT]umontreal[DOT]ca
ago109[AT]psu[DOT]edu
jpineau[AT]cs[DOT]mcgill[DOT]ca
aaron[DOT]courville[AT]umontreal[DOT]ca

## Abstract

Recent advances in neural variational inference have facilitated efficient training of powerful directed graphical models with continuous latent variables, such as variational autoencoders. However, these models usually assume simple, uni-modal priors — such as the multivariate Gaussian distribution — yet many real-world data distributions are highly complex and multi-modal. Examples of complex and multi-modal distributions range from topics in newswire text to conversational dialogue responses. When such latent variable models are applied to these domains, the restriction of the simple, uni-modal prior hinders the overall expressivity of the learned model as it cannot possibly capture more complex aspects of the data distribution. To overcome this critical restriction, we propose a flexible, simple prior distribution which can be learned efficiently and potentially capture an exponential number of modes of a target distribution. We develop the multi-modal variational encoder-decoder framework and investigate the effectiveness of the proposed prior in several natural language processing modeling tasks, including document modeling and dialogue modeling.

## 1 Introduction

With the development of the variational autoencoding framework (Kingma & Welling, 2013; Rezende et al., 2014), a tremendous amount of progress has been made in learning large-scale, directed latent variable models. This approach has lead to improved performance in applications ranging from computer vision (Gregor et al., 2015; Larsen et al., 2015) to natural language processing (Mnih & Gregor, 2014; Miao et al., 2015; Bowman et al., 2015; Serban et al., 2016b). Furthermore, these models naturally incorporate a Bayesian modeling perspective, by enabling the integration of problem-dependent knowledge in the form of a prior on the generating distribution.

However, the majority of models proposed assume an extremely simple prior in the form of a multivariate Gaussian distribution in order to maintain mathematical and computational tractability. Although this assumption on the prior has lead to favorable results on several tasks, it is clearly a restrictive and often unrealistic assumption. First, it imposes a strong uni-modal structure on the latent variable space; latent samples from the generating model (prior distribution) all cluster around a single mean. Second, it encourages local smoothness on the latent variables; the similarity between two latent variables decreases exponentially as their distance increase. Thus, for complex, multi-modal distributions — such as the distribution over topics in a text corpus, or natural language responses in a dialogue system — the uni-modal Gaussian prior inhibits the model's ability to extract and represent important structure in the data. To learn more powerful and expressive models — in particular, models with multi-modal latent variable structures for natural language processing applications — we seek a suitable and flexible prior than can be automatically adapted to model multiple modes of a target distribution.

---

[*]First two authors contributed equally.

In this paper, we propose the multi-modal variational encoder-decoder framework, introducing an efficient, flexible prior distribution that is suitable for distributions such as those found in natural language text. We demonstrate the effectiveness of our multi-modal variational architectures in two representative tasks: document modeling and dialogue modeling. We find that our prior is able to capture elements of a target distribution that simpler priors — such as the uni-modal Gaussian — cannot model, thus allowing neural latent variable models to extract richer structure from data. In particular, we achieve state-of-the-art results on several document modeling tasks.

## 2 RELATED WORK

The idea of using an artificial neural network to approximate an inference model dates back to the 90s (Hinton & Zemel, 1994; Hinton et al., 1995; Dayan & Hinton, 1996). However, initial attempts at such an approach were hindered by the lack of low-bias, low-variance estimators of parameter gradients. Traditionally, researchers resorted to Markov chain Monte Carlo methods (MCMC) (Neal, 1992) which do not scale well and mix slowly, or to variational approaches which require a tractable, factored distribution to approximate the true posterior distribution, usually under-fitting it (Jordan et al., 1999). Others have since proposed using feed-forward inference models to efficiently initialize the mean-field inference algorithm for incrementally training Boltzmann architectures (Salakhutdinov & Larochelle, 2010; Ororbia II et al., 2015b). However, these approaches are limited by the mean-field inference's inability to model structured posteriors. Recently, Mnih & Gregor (2014) proposed the neural variational inference and learning (NVIL) approach to match the true posterior directly without resorting to approximate inference. NVIL allows for the joint training of an inference network and directed generative model, maximizing a variational lower-bound on the data log-likelihood and facilitating exact sampling of the variational posterior. Simultaneously with this work, the variational autoencoder framework was proposed by Kingma & Welling (2013) and Rezende et al. (2014). This framework is the motivation of this paper, and will be discussed in detail in the next section.

With respect to document modeling, it has recently been demonstrated that neural architectures can outperform well-established, standard topic models such as Latent Dirichlet Allocation (LDA) (Blei et al., 2003). For example, it has been demonstrated that models based on the Boltzmann machine, which learn semantic binary vectors (binary latent variables), perform very well (Hofmann, 1999). Work involving discrete latent variables include the constrained Poisson model (Salakhutdinov & Hinton, 2009), the Replicated Softmax model (Hinton & Salakhutdinov, 2009) and the Over-Replicated Softmax model (Srivastava et al., 2013), as well as similar, auto-regressive neural architectures and deep directed graphical models (Larochelle & Lauly, 2012; Uria et al., 2014; Lauly et al., 2016; Bornschein & Bengio, 2014). In particular, Mnih & Gregor (2014) showed that using NVIL yields better generative models of documents than these previous approaches. The success of these discrete latent variable models — which are able to partition probability mass into separate regions — serve as the main motivation for investigating models with continuous multi-modal latent variables for document modeling. More recently, Miao et al. (2015) have proposed continuous latent variable representations for document modeling, which has achieved state-of-the-art results. This model will be described later.

With respect to dialogue modeling, latent variable models were investigated by Bangalore et al. (2008), Crook et al. (2009) as well as others. More recently, Zhai & Williams (2014) have proposed three models combining hidden Markov models and topic models. The success of these discrete latent variable models also motivates our investigation into dialogue models with multi-modal latent variables. Most related to our work is the Variational Hierarchical Recurrent Encoder-Decoder (VHRED) model by Serban et al. (2016b), which is a neural architecture with latent multivariate Gaussian variables. This model will be described later.

There has been some work exploring alternative distributions for the latent variables in the variational autoencoder framework, including multi-modal distributions. Rezende & Mohamed (2015) propose an approach called normalizing flows which computes a more complex, potentially multi-modal distribution, by projecting standard Gaussian variables through a sequence of non-linear transformations. This approach is similar to the inverse auto-regressive flow proposed by Kingma et al. (2016). Unfortunately, both normalizing flows and auto-regressive flow are only applicable to the approximate posterior distribution; typically these approaches require fixing the prior distri-

bution to a uni-modal multivariate Gaussian. Furthermore, to the best of our knowledge, neither of these approaches have been investigated in the context of larger scale text processing tasks, such as the document modeling and dialogue modeling tasks we evaluate on. A complementary approach is to combine variational inference with MCMC sampling (Salimans et al., 2015; Burda et al., 2015), however this is computationally expensive and therefore difficult to scale up to many real-world tasks. Enriching the latent variable distributions has also been investigated by Maaløe et al. (2016).

## 2.1 APPROACHES FOR LEARNING MULTI-MODAL LATENT VARIABLES

**Mixture of Gaussians** Perhaps the most direct and naive approach to learning multi-modal latent variables is to parametrize the latent variable prior and approximate posterior distributions as a mixture of Gaussians. However, the KL divergence between two mixtures of Gaussian distributions cannot be computed in closed form (Durrieu et al., 2012). To train such a model, one would have to either resort to MCMC sampling, which may slow down and hurt the training process due to the high variance it incurs, or resort to approximations of the KL divergence, which may also hurt the training process.[1]

**Deep Directed Models** An alternative to a mixture of Gaussians parametrization is to construct a deep directed graphical model composed of multiple layers of uni-modal latent variables (e.g. multivariate Gaussians) (Rezende et al., 2014). Such models have the potential to capture highly complex, multi-modal latent variable representations through the marginal distribution of the top-layer latent variables. However, this approaches has two major drawbacks. First, the variance of the gradient estimator grows with the number of layers. This makes it difficult to learn highly multi-modal latent representations. Second, it is not clear how many modes such models can represent or how their inductive biases will affect their performance on tasks containing multi-modal latent structure. The piecewise constant latent variables we propose do not suffer from either of these two drawbacks; the piecewise constant variables incur low variance in the gradient estimator, and can, in principle, represent a number of modes exponential in the number of latent variables.

**Discrete Latent Variables** A third approach for learning multi-modal latent representations is to instead use discrete latent variables as discussed above. For example, the learning procedure proposed by Mnih & Gregor (2014) for discrete latent variables can easily be combined with the variational autoencoder framework to learn models with both discrete and continuous latent variables. However, the major drawback of discrete latent variables is the high variance in the gradient estimator. Without further approximations, it might be difficult to scale up models with discrete latent variables for real-world tasks.

## 3 THE MULTI-MODAL VARIATIONAL ENCODER-DECODER FRAMEWORK

We start by describing the general neural variational learning framework. Then we present our proposed prior model aimed at enhancing the model's ability to learn multiple modes of data distributions. We focus on modeling discrete output variables in the context of natural language processing applications. However, the framework can easily be adapted to handle continuous output variables, such as images, video and audio.

## 3.1 NEURAL VARIATIONAL LEARNING

Let $w_1, \ldots, w_N$ be a sequence of $N$ words conditioned on a continuous latent variable $z$. In the general framework, the distribution over the variables follows the directed graphical model:

$$P_\theta(w_1, \ldots, w_N, z) = \int \prod_{n=1}^{N} P_\theta(w_n | w_{<n}, z) P_\theta(z) dz, \tag{1}$$

where $\theta$ are the model parameters. The model first generates the higher-level, continuous latent variable $z$, and then, conditioned on this, generates the word sequence. The document modeling

---

[1]Our lab has previously investigated incorporating mixture of Gaussian models into the autoencoder framework, but without any success. This work has not been published.

task further simplifies the model by assuming the words are independent of each other:

$$P_\theta(w_1, \ldots, w_N, z) = \int \prod_{n=1}^{N} P_\theta(w_n|z) P_\theta(z) dz. \tag{2}$$

Following the variational autoencoder (VAE) framework (Kingma & Welling, 2013), the parameters can be learned using the variational lower-bound:

$$\log P_\theta(w_1, \ldots, w_N, z) \geq \mathrm{E}_{z \sim Q_\psi(z|w_1, \ldots, w_N)}[\log P_\theta(w_n|w_{<n}, z)] - \mathrm{KL}\left[Q_\psi(z|w_1, \ldots, w_N) || P_\theta(z)\right], \tag{3}$$

where $Q_\psi(z|w_1, \ldots, w_N)$ is the approximation to the posterior for $z$, called the *encoder*, or sometimes the *recognition model* or *inference model*, with parameters $\psi$. The distribution $P_\theta(z)$ is the prior model for $z$. The variational autoencoder model further makes use of the re-parametrization trick, which allows one to move the derivative of the lower-bound to inside the expectation. To accomplish this, we need to parametrize $z$ as a transformation from a fixed (parameter-less) random distribution:

$$z = f_\theta(\epsilon), \tag{4}$$

where $\epsilon$ is drawn from a random distribution, e.g. a standard Gaussian distribution (with zero mean and unit standard deviation) or a uniform distribution in the interval $[0, 1]$, and $f$ is some transformation of this variable, also parametrized by $\theta$.

The majority of work on VAEs that uses the re-parametrization trick propose to parametrize $z$ — both the prior and approximate posterior (encoder) — as a multivariate Gaussian variable. However, the multivariate Gaussian is a uni-modal distribution and can therefore only represent one mode in latent space. This means the mapping from latent variable to outputs — i.e. the conditional distribution $P_\theta(w_n|z)$ — has to be highly non-linear in order to capture additional modes. However, in general, it is difficult to learn such non-linear mappings with existing stochastic optimization methods, such as mini-batch stochastic gradient descent and its variants. Learning such a non-linear mapping is particularly difficult using the variational bound in eq. (3), because it incurs additional variance from sampling the latent variable $z$. Consequently, such a model is very likely to converge on a solution which does not model multi-modality which then leads to a poor approximation of the output distribution.

## 3.2 THE PIECEWISE-CONSTANT PRIOR FOR LATENT VARIABLES

In this work, we overcome the uni-modal restriction by parametrizing $z$ using a piecewise constant probability density function (PDF). This parametrization will allow $z$ to represent complex aspects of the data distribution in latent variable space, such as multiple modes and highly non-smooth regions of probability mass. From a manifold learning perspective, this extension translates into expanding the set of manifolds representable by the model parameters to include more non-linear manifolds – in particular, manifolds where there exists separate clusters of probability mass.

Let $n \in \mathbb{N}$ be the number of piecewise constant components. We assume $z$ is drawn from the PDF:

$$P(z) = \frac{1}{K} \sum_{i=1}^{n} 1\left(\frac{i-1}{n} \leq z \leq \frac{i}{n}\right) a_i, \tag{5}$$

where $1_{(x)}$ is the indicator function (which is one whenever $x$ is true and otherwise zero), $a_i > 0$ for $i = 1, \ldots, n$ are the distribution parameters (which will be learned during training), and $K$ is the normalization constant:

$$K = \sum_{i=1}^{n} K_i, \quad \text{where } K_0 := 0, K_i := \frac{a_i}{n} \text{ for } i = 1, \ldots, n. \tag{6}$$

To train the model using the re-parametrization trick, we need to generate $z = f(\epsilon)$ where $\epsilon \sim \text{Uniform}(0, 1)$. To do so, we employ inverse transform sampling (Devroye, 1986), which requires finding the inverse of the cumulative distribution function (CDF). We first derive the CDF of eq. (5):

$$\phi(z) = \frac{1}{K} \sum_{i=1}^{n} 1\left(\frac{i}{n} \leq z\right) K_i + 1\left(\frac{i-1}{n} \leq z \leq \frac{i}{n}\right) \left(z - \frac{i-1}{n}\right) a_i. \tag{7}$$

Next, we derive its inverse:

$$\phi^{-1}(\epsilon) = \sum_{i=1}^{n} 1 \left( \frac{1}{K} \sum_{j=0}^{i-1} K_j \leq \epsilon \leq \frac{1}{K} \sum_{j=0}^{i} K_j \right) \left( \frac{i-1}{n} + \frac{K}{a_i} \left( \epsilon - \frac{1}{K} \sum_{j=0}^{i-1} K_j \right) \right) \quad (8)$$

Armed with the inverse CDF, we can now draw a sample $z$:

$$z = \phi^{-1}(\epsilon), \quad \text{where } \epsilon \sim \text{Uniform}(0, 1). \quad (9)$$

In addition to sampling, we need to compute the Kullback-Leibler (KL) divergence between the prior and approximate posterior distributions of the piecewise constant variables. We assume both the prior and the posterior are piecewise constant distributions. We use *prior* to denote prior parameters and *post* to denote posterior parameters (encoder model parameters). The KL divergence between the prior and posterior can be computed using a sum of integrals, where each integral inside the sum corresponds to one constant segment:

$$\text{KL}\left[Q_\psi(z|w_1, \ldots, w_N)||P_\theta(z)\right] = \int_0^1 Q_\psi(z|w_1, \ldots, w_N) \log \left( \frac{Q_\psi(z|w_1, \ldots, w_N)}{P_\theta(z)} \right) dz \quad (10)$$

$$= \sum_{i=1}^{n} \int_0^{1/n} \frac{a_i^{\text{post}}}{K^{\text{post}}} \log \left( \frac{a_i^{\text{post}}/K^{\text{post}}}{a_i^{\text{prior}}/K^{\text{prior}}} \right) dz \quad (11)$$

$$\quad (12)$$

$$= \frac{1}{n} \sum_{i=1}^{n} \frac{a_i^{\text{post}}}{K^{\text{post}}} \log \left( \frac{a_i^{\text{post}}/K^{\text{post}}}{a_i^{\text{prior}}/K^{\text{prior}}} \right) \quad (13)$$

$$= \frac{1}{n} \frac{1}{K^{\text{post}}} \sum_{i=1}^{n} a_i^{\text{post}} \left( \log(a_i^{\text{post}}) - \log(a_i^{\text{prior}}) \right) \quad (14)$$

$$+ \log(K^{\text{prior}}) - \log(K^{\text{post}})$$

In order to train the model, we take partial derivatives of the variational bound in eq. (3) w.r.t. each parameter in $\theta$ and $\psi$. These expressions involve derivatives of the indicator functions, which have derivatives zero everywhere except for the changing points where the derivative is undefined. However, the probability of sampling $\epsilon$ such that an indicator function is exactly at its changing point is effectively zero. Therefore, we fix their derivatives to zero.[2] A similar approach is used for training neural networks with rectified linear units. Figure 1 illustrates how the piecewise constant latent variables can work with Gaussian latent variables in order to model multi-modality.

## 4 LATENT VARIABLE PARAMETRIZATIONS

The latent variable parametrizations are crucial to modeling the data effectively. In this section, we will develop the parametrizations for both the Gaussian variable and our proposed piecewise latent variable.

For all parametrizations, let $c$ be the conditioning information for the prior. In document modeling there is no conditioning information available to the prior, so $c = \emptyset$. In dialogue modeling $c$ is the vector representation of the dialogue context, namely all previous utterances until the current time step. Let $x$ be the current output sequence (observation), which the model must generate (e.g. $w_1, \ldots, w_N$ for document modeling).

### 4.1 GAUSSIAN PARAMETRIZATION

Let $\mu^{\text{prior}}$ and $\sigma^{2,\text{prior}}$ be the prior mean and variance, and let $\mu^{\text{post}}$ and $\sigma^{2,\text{post}}$ be the posterior mean and variance. For Gaussian latent variables, the prior distribution mean and variances are encoded using linear transformations of a hidden state. In particular, the prior distribution covariance is

---

[2]We thank Christian A. Naesseth for pointing out this assumption.

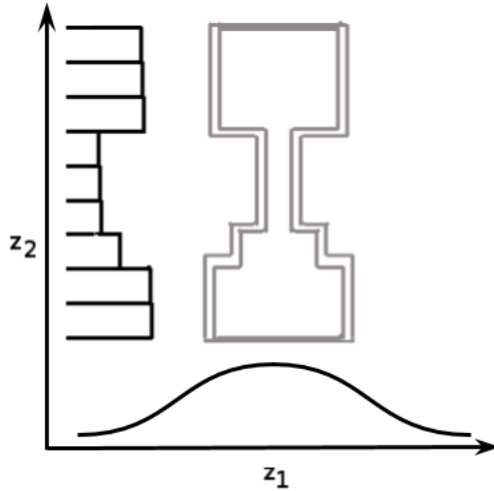

Figure 1: The horizontal axis corresponds to $z_1$, which is a univariate Gaussian variable. The vertical axis corresponds to $z_2$, which is a piecewise constant variable. The PDF for each variable is shown along each axis, and their joint distribution is illustrated in grey color.

encoded as a diagonal covariance matrix using a softplus function:

$$\mu^{\text{prior}} = H_\mu^{\text{prior}}\text{Enc}(c) + b_\mu^{\text{prior}}, \tag{15}$$

$$\sigma^{2,\text{prior}} = \text{diag}(\log(1 + \exp(H_\sigma^{\text{prior}}\text{Enc}(c) + b_\sigma^{\text{prior}}))), \tag{16}$$

where $\text{Enc}(c)$ is an embedding/encoding of the context $c$ (e.g. given by a bag-of-words encoder or an LSTM encoder applied to $c$), which is shared across all latent variable dimensions. The parameters $H_\mu^{\text{prior}}, b_\mu^{\text{prior}}, H_\sigma^{\text{prior}}, b_\sigma^{\text{prior}}$ are to be learned.

For the posterior distribution, our preliminary experiments have shown that it is much better to parametrize the posterior distribution by interpolating between the prior distribution mean and variance and a new estimate of the mean and variance. This interpolation is controlled by a gating mechanism, which makes it easy for the model to learn how to turn on/off latent dimensions:

$$\mu^{\text{post}} = (1 - \alpha_\mu)\mu^{\text{prior}} + \alpha_\mu \left( H_\mu^{\text{post}}\text{Enc}(c, x) + b_\mu^{\text{post}} \right), \tag{17}$$

$$\sigma^{2,\text{post}} = (1 - \alpha_\sigma)\sigma^{2,\text{prior}} + \alpha_\sigma\text{diag}(\log(1 + \exp(H_\sigma^{\text{post}}\text{Enc}(c, x) + b_\sigma^{\text{post}}))), \tag{18}$$

where $\text{Enc}(c, x)$ is an encoding/embedding of both $c$ and $x$, and where the parameters are $H_\mu^{\text{post}}, b_\mu^{\text{post}}, H_\sigma^{\text{post}}, b_\sigma^{\text{post}}, \alpha_\mu, \alpha_\sigma$. The interpolation mechanism is controlled by $\alpha_\mu$ and $\alpha_\sigma$, which are initialized to zero (i.e. initialized such that the posterior is equal to the prior).[3]

### 4.2 Piecewise Constant Parametrization

Similar to the Gaussian variances, we propose to parametrize the piecewise constant prior parameters using an exponential function applied to a linear transformation of the context embedding/encoding:

$$a_i^{\text{prior}} = \exp(H_{a,i}^{\text{prior}}\text{Enc}(c) + b_{a,i}^{\text{prior}}), \quad i = 1, \ldots, n, \tag{19}$$

where $H_a^{\text{prior}}$ and $b_a^{\text{prior}}$ are the parameters to be learned.

We may also constrain the piecewise constant posterior parameters to be an interpolation between the prior parameters and a new estimated parameter:

$$a_i^{\text{post}} = (1 - \alpha_{a,i})a_i^{\text{prior}} + \alpha_{a,i}\exp(H_{a,i}^{\text{post}}\text{Enc}(c, x) + b_{a,i}^{\text{post}}), \quad i = 1, \ldots, n, \tag{20}$$

---

[3]We experimented with more sophisticated mechanisms for controlling the gating variables, including defining $\alpha_\mu$ and $\alpha_\sigma$ to be a linear function of the encoder. However, we found that simpler was often better and thus do not report these results using more advanced mechanisms.

where $H_a^{\text{post}}, b_a^{\text{post}}, \alpha_a$ are the parameters. However, we found that this interpolation hurt performance and therefore fixed $\alpha_a = 1$.

To take advantage of the properties of both priors, the Gaussian and piecewise constant variables may be combined, as was suggested in Section 3.2. In this work, we primarily experimented with their concatenation to create a hybrid model.

# 5   VARIATIONAL TEXT MODELING

We now present two probabilistic models, the NVDM and the VHRED, which are extended to incorporate the latent variable parametrization and used for the document modeling and the dialogue modeling experiments described below.

## 5.1   NEURAL VARIATIONAL DOCUMENT MODEL (NVDM)

The NVDM framework (Mnih & Gregor, 2014; Miao et al., 2015) collapses the recurrent neural encoder into a simpler bag-of-words model (since no symbol order is taken into account), which may be defined as a multi-layer perceptron (MLP) for $Enc(c = \emptyset, x) = Enc(x)$. Let $V$ be the vocabulary. Let $W$ represent a document matrix, where row $w_i$ is the 1-of-$|V|$ binary encoding of the $i$'th word in the document. $Enc(W)$ is trained to compress a document vector into a continuous distributed representation upon which the posterior model is built.

The NVDM parametrization requires only learning the parameters $b_a^{\text{prior}}, W_a^{\text{post}}, b_a^{\text{post}}$ for the piecewise variables, and learning the parameters $b_\mu^{\text{prior}}, b_\sigma^{\text{prior}}, W_\mu^{\text{post}}, b_\mu^{\text{post}}, W_\sigma^{\text{post}}, b_\sigma^{\text{post}}$ for the Gaussian variables. We initialize the bias parameters to zero, in order for the NVDM to start with a centered Gaussian prior. This prior will be adapted by the parametric encoder as learning progresses, while also learning to turn on/off latent dimensions controlled through the gating mechanism. It is important to note that our particular instantiation of the NVDM is different from that of Mnih & Gregor (2014) and Miao et al. (2015); we jointly learn the prior mean and variance whereas in previous work it has been assumed to be a standard Gaussian. Furthermore, our models learn to interpolate between the generated prior and posterior models to calculate a new posterior.

Based on preliminary experiments, we choose the encoder to be a 2-hidden layer perceptron, defined by parameters $\{E^0, b^0, E^1, b^1\}$. The decoder is defined by parameters $\{R, c\}$. For example, in the case of the hybrid VAE we use eq. (15)–(20) to generate the distribution parameters. In this case, to draw a sample from the Gaussian prior, we draw a standard Gaussian variable and then multiply it by the standard deviation and add the mean of the Gaussian prior. To draw a sample from the piecewise prior, we use eq. (8). As such, the complete architecture is:

$$\pi(W) = f^0(E^0 W + b^0),$$
$$Enc(W) = f^1(E^1 \pi(W) + b^1),$$
$$z_{Gaussian} = \mu^{\text{post}} + \sqrt{\sigma^{2,\text{post}}} \otimes \epsilon_0,$$
$$z_{Piecewise} = \phi^{-1,post}(\epsilon_1),$$
$$z = \langle z_{Gaussian}, z_{Piecewise} \rangle,$$
$$Dec(w, z) = g(-w^{\mathrm{T}} R z),$$

where $\otimes$ is the Hadamard product, $\langle \circ, \circ \rangle$ is an operator that combines the Gaussian and the Piecewise variables and $Dec(w, z)$ is the decoder model. [4] As a result of using the re-parametrization trick and choice of prior, we calculate the latent variable $z$ through the two samples, $\epsilon_0$ and $\epsilon_1$. $f(\circ)$ is a non-linear activation function. We choose it to be the softsign function, or $f(v) = v/(1 + |v|)$. The decoder model $Dec(z)$ outputs a probability distribution over words conditioned on $z$. In this case, we define $g(\circ)$ as the softmax function (omitting the bias term $c$ for clarity) computed as:

$$Dec(w, z) = P_\theta(w|z) = \frac{\exp\left(-w^{\mathrm{T}} R z\right)}{\sum_{w'} \exp\left(-w^{\mathrm{T}} R z\right)},$$

---

[4]Operations include vector concatenation, summation, or averaging.

The decoder's output is used to calculate the first term in the variational lower-bound: $\log P_\theta(W|z)$. The prior and posterior distributions are used to compute the KL term in the variational lower-bound. The lower-bound defined becomes:

$$\mathcal{L} = \mathrm{E}_{Q_\psi(z|W)}\left[\sum_{i=1}^{N}\log P_\theta(w_i|z)\right] - \mathrm{KL}\left[Q_\psi(z|W)||P_\theta(z)\right],$$

where the KL term is the sum of the Gaussian and piecewise KL-divergence measures:

$$\mathrm{KL}\left[Q(z|W)||P(z)\right] = \mathrm{KL}_{Gaussian}\left[Q(z|W)||P(z)\right] + \mathrm{KL}_{Piecewise}\left[Q(z|W)||P(z)\right].$$

The KL-terms may be interpreted as regularizers of the parameter updates for the encoder model (Kingma & Welling, 2013). These terms encourage the posterior distributions to be similar to their corresponding prior distributions, by limiting the amount of information the encoder model transmits regarding the output. For example, it encourages the uni-modal Gaussian posterior to move its mean close to the mean of the Gaussian prior, which makes it difficult for the Gaussian posterior to represent different modes conditioned on the observation. Similarly, this encourages the piecewise constant posterior to be similar to the piecewise constant prior. However, since the piecewise constant posterior is multi-modal, it may be able to shift some of its probability mass towards the prior distribution while keeping other probability mass on one or several modes dependent upon the output observation (e.g. if the prior distribution is a uniform distribution and the true posterior concentrates all its probability mass in several small regions, then the approximate posterior could interpolate between the prior and the true posterior).

## 5.2 VARIATIONAL HIERARCHICAL RECURRENT ENCODER-DECODER (VHRED)

The VHRED model is an extension of the hierarchical recurrent encoder-decoder model (HRED) for dialogue (Serban et al., 2016b;a). The model decomposes dialogues using a two-level hierarchy: sequences of utterances (e.g. sentences), and sub-sequences of tokens (words). Let $\mathbf{w}_n$ be the $n$'th utterance in a dialogue with $N$ utterances. Let $w_{n,m}$ be the $m$'th word in the $n$'th utterance from vocabulary $V$, and let $M_n$ be the number of words in the $n$'th utterance. In addition to this, VHRED has a latent multivariate continuous variable $\mathbf{z}_n$ for each utterance $n = 1, \ldots, N$. The probability distribution of the generative model factorizes as:

$$P_\theta(\mathbf{w}_1, \ldots, \mathbf{w}_N) = \prod_{n=1}^{N} P_\theta(\mathbf{w}_n|\mathbf{w}_{<n}, \mathbf{z}_n)P_\theta(\mathbf{z}_n|\mathbf{w}_{<n}),$$

$$= \prod_{n=1}^{N}\prod_{m=1}^{M_n} P_\theta(w_{n,m}|w_{n,<m}, \mathbf{w}_{<n}, \mathbf{z}_n)P_\theta(\mathbf{z}_n|\mathbf{w}_{<n}), \qquad (21)$$

where $\theta$ are the model parameters. VHRED uses three RNN modules: an *encoder* RNN, a *context* RNN and a *decoder* RNN. First, each utterance is encoded into a vector by the *encoder* RNN:

$$h_{n,0}^{\mathrm{enc}} = \mathbf{0}, \quad h_{n,m}^{\mathrm{enc}} = f_\theta^{\mathrm{enc}}(h_{n,m-1}^{\mathrm{enc}}, w_{n,m}) \,\forall m = 1, \ldots, M_n,$$

where $f_\theta^{\mathrm{enc}}$ is either a GRU or a bidirectional GRU function. The last hidden state of the *encoder* RNN is given as input to the *context* RNN. Then, the *context* RNN updates its internal hidden state to reflect all the information up until that utterance:

$$h_0^{\mathrm{con}} = \mathbf{0}, \quad h_n^{\mathrm{con}} = f_\theta^{\mathrm{con}}(h_{n-1}^{\mathrm{con}}, h_{n,M_n}^{\mathrm{enc}}),$$

where $f_\theta^{\mathrm{con}}$ is a GRU function taking as input two vectors. This state is used to compute the prior distribution over the latent variable $\mathbf{z}_n$:

$$P_\theta(\mathbf{z}_n \mid \mathbf{w}_{<n}) = f_\theta^{\mathrm{prior}}(h_{n-1}^{con}), \qquad (22)$$

where $f^{\mathrm{prior}}$ is a PDF parametrized by both $\theta$ and $h_n^{con}$. Next, a sample is drawn from this distribution: $\mathbf{z}_n \sim P_\theta(\mathbf{z}_n|\mathbf{w}_{<n})$. The sample and *context* state are given as input to the *decoder* RNN:

$$h_{n,0}^{\mathrm{dec}} = \mathbf{0}, \quad h_{n,m}^{\mathrm{dec}} = f_\theta^{\mathrm{dec}}(h_{n,m-1}^{\mathrm{dec}}, h_{n-1}^{\mathrm{con}}, \mathbf{z}_n, w_{n,m})$$
$$\forall m = 1, \ldots, M_n,$$

where $f_\theta^{\text{dec}}$ is the LSTM gating function taking as input four vectors. The output distribution is computed by passing $h_{n,m}^{\text{dec}}$ through an MLP $f_\theta^{\text{mlp}}$, an affine transformation and a softmax function:

$$P_\theta(w_{n,m+1}|w_{n,\leq m}, \mathbf{w}_{<n}, \mathbf{z}_n) = \frac{e^{(Ow_{n,m+1})^{\text{T}} f_\theta^{\text{mlp}}(h_{n,m}^{\text{dec}})}}{\sum_{w'} e^{(Ow')^{\text{T}} f_\theta^{\text{mlp}}(h_{n,m}^{\text{dec}})}}, \qquad (23)$$

where $O \in \mathbb{R}^{|V| \times d}$ is the word embedding matrix for the output distribution with embedding dimensionality $d \in \mathbb{N}$. The model is trained by maximizing the variational lower-bound, which factorizes into independent terms for each sub-sequence (utterance):

$$\log P_\theta(\mathbf{w}_1, \ldots, \mathbf{w}_N) \geq \sum_{n=1}^{N} - \text{KL}\left[Q_\psi(\mathbf{z}_n \mid \mathbf{w}_1, \ldots, \mathbf{w}_n)||P_\theta(\mathbf{z}_n \mid \mathbf{w}_{<n})\right]$$
$$+ \mathbb{E}_{Q_\psi(\mathbf{z}_n|\mathbf{w}_1,\ldots,\mathbf{w}_n)}\left[\log P_\theta(\mathbf{w}_n \mid \mathbf{z}_n, \mathbf{w}_{<n})\right], \qquad (24)$$

where distribution $Q_\psi$ is the approximate posterior distribution with parameters $\psi$, which is computed similar to the prior distribution but further conditioned on the future *encoder* RNN hidden state:

$$Q_\psi(\mathbf{z}_n \mid \mathbf{w}_{\leq n}) = f_\psi^{\text{post}}(h_{n-1}^{con}, h_{n,M_n}^{enc}), \qquad (25)$$

where $f^{\text{post}}$ is a PDF. More details are given by Serban et al. (2016b).

The original VHRED model as described by Serban et al. (2016b) used only Gaussian latent variables. We will refer to this model as Gaussian-VHRED (G-VHRED). The VHRED model with both Gaussian and piecewise constant latent variables will be referred to as Hybrid-VHRED (H-VHRED). In this case, we combine the Gaussian and piecewise latent variables by concatenating them into one vector.[5]

## 6 EXPERIMENTS

In order to validate the ability of our piecewise latent variables to capture complex aspects of data distributions, we conduct experiments with both the NVDM and VHRED models.

All models are trained using back-propagation to obtain parameter gradients with respect to the variational lower-bound on the log-likelihood or the exact log-likelihood. We used a standard first-order gradient-descent optimizer, Adam (Kingma & Ba, 2015), for both models, where only hyperparameter choices varied depending on the task. The specifics of the design of the encoder and decoder differed between the two tasks (as described in Sections 5.1 and 5.2). For all models that used piecewise latent variables, we chose to fix $\alpha_{a_i} = 1$, meaning the piecewise prior and posterior models are kept separate (instead of having the posterior be an interpolation between another distribution and the prior), since we found this to perform better[6]

### 6.1 DOCUMENT MODELING

For our experiments in document modeling, we make use of the 20 News-Groups dataset. We follow the pre-processing and set-up of Hinton & Salakhutdinov (2009). In addition, we make use of the Reuters corpus (RCV1-V2), using a version that contained a selected 5,000 term vocabulary. [7] Note that the features are a $\log(1 + TF)$ transform of the original frequency vectors. To test our document models on text from another language (in this case, Brazilian Portuguese), we make use of the CADE12 dataset (stop-word removed and stemmed) Cardoso-Cachopo (2007), where we further filtered terms that occurred less than 130 times to obtain a vocabulary of 3,736 terms (over 26,991 training and 13,486 test documents). For all datasets, we track the validation bound on a subset of 100 vectors randomly drawn from each training corpus.

---

[5]Before concatenation, we transform the piecewise constant latent variables to lie within the interval $[-1, 1]$: $\mathbf{z}' = 2\mathbf{z} - 1$. This ensures the input to the decoder RNN has mean zero at the beginning of training.

[6]We believe that if $\alpha_{a_i} = 0$ for a long period of time, then the posterior receives no gradient signal. Without a gradient signal, the estimated posterior becomes increasingly disconnected from the rest of the model and, thus, less effective. This might be due to the choice of non-linearities, which affect the piecewise latent variables moreso than the Gaussian latent variables.

[7]We will make the code and scripts used to create the final document input vectors and vocabulary files publicly available upon publication.

| 20-NG | Sampled | SGD-Inf |
|---|---|---|
| *LDA* | 1058 | −− |
| *RSM* | 953 | −− |
| *docNADE* | 896 | −− |
| *SBN* | 909 | −− |
| *fDARN* | 917 | −− |
| *NVDM* | 836 | −− |
| *G-NVDM* | 651 | 588 |
| *H-NVDM-3* | 607 | 546 |
| *H-NVDM-5* | **566** | **496** |

| RCV1 | Sampled | SGD-Inf |
|---|---|---|
| *G-NVDM* | 905 | 837 |
| *H-NVDM-3* | 865 | 807 |
| *H-NVDM-5* | **833** | **781** |

| CADE | Sampled | SGD-Inf |
|---|---|---|
| *G-NVDM* | 339 | 230 |
| *H-NVDM-3* | **258** | **193** |
| *H-NVDM-5* | 294 | 209 |

Table 1: Comparative test perplexities on various document datasets (50 latent variables). Note that document probabilities were calculated using 10 samples to estimate the variational lower bound.

| G-NVDM | H-NVDM-3 | H-NVDM-5 | G-NVDM | H-NVDM-3 | H-NVDM-5 |
|---|---|---|---|---|---|
| governments | citizens | arms | environment | project | science |
| citizens | rights | rights | project | gov | built |
| country | governments | federal | flight | major | high |
| threat | civil | country | lab | based | technology |
| private | freedom | policy | mission | earth | world |
| rights | legitimate | administration | launch | include | form |
| individuals | constitution | protect | field | science | scale |
| military | private | private | working | nasa | sun |
| freedom | court | citizens | build | systems | special |
| foreign | states | military | gov | technical | area |

Table 2: Word query similarity test, where each (20 News-Group) document model's decoder is given a query and must return the top 10 most relevant words. The first query was "government" while the second was "space". It appears that the models with piecewise variables tend to associate more general/abstract terms to the query, which may or may not always be what is desired.

For the Gaussian NVDM (*G-NVDM*), we constrain the interpolated posterior variance to lie in the range of $[0.01, 10.0]$. For the hybrid NVDMs (*H-NVDM*) [8], we vary the number of components used in the PDF, investigating the effect that 3 and 5 pieces had on the final quality of the model. Parameter updates for all models were estimated using mini-batches of 100 samples drawn randomly without replacement from the training data over 150 epochs. A learning rate of 0.002 was used. Model selection and early stopping (the only additional form of regularization employed for this set of experiments) were conducted using the validation lower-bound, estimated using five stochastic samples per validation example. We rescale large gradients by their norm (Pascanu et al., 2012). Inference networks made use of 50 units in each hidden layer for 20 News-Groups and CADE and 100 for RCV1, while all performed best with 50 latent variables (chosen via preliminary experimentation with smaller models). On the 20 News-Groups, since we were able to use the same set-up (especially vocabulary) as Hinton & Salakhutdinov (2009), we also report the perplexities of a topic model (*LDA*, Hinton & Salakhutdinov (2009)), the Replicated Softmax (*RSM*, Hinton & Salakhutdinov (2009)), the document neural auto-regressive estimator (*docNADE*, Larochelle & Lauly (2012)), a sigmoid belief network (*SBN*, Mnih & Gregor (2014)), a deep auto-regressive neural network (*fDARN*, Mnih & Gregor (2014)), and a neural variational document model with a fixed standard Gaussian prior (*NVDM*, lowest reported perplexity, Miao et al. (2015)).

In Table 1, we report the test document perplexity (under the *Sampled* column), calculated using the standard formula, $exp(-\frac{1}{D} \sum_n \frac{1}{L_n} \log P_\theta(x_n))$. Note that $\log P_\theta(x_n)$, or the log-probability of a particular document, was approximated with an estimate of the variational lower-bound using 10 samples, as was done in Mnih & Gregor (2014). The second score (or column *SGD-Inf*), refers to the model's test-perplexity when the lower-bound is tightened using iterative inference to search for the optimal latent variable per document. In this paper, our iterative inference procedure consisted of simple stochastic gradient descent (no more than 100 steps), with a learning rate of 0.1 and the same

---

[8]We ultimately found that averaging the variables, as opposed to using concatenation, yielded best perplexity and thus report these results.

gradient rescaling used in training, using early-stopping (for 20 News-Groups, the lookahead was 10, while on Reuters and CADE the lookahead was 5). The parameters of the model, as well as the well as the generated prior, are fixed, and the gradients of the variational lower bound with respect to generated posterior model parameters (i.e., the mean and variance of the Gaussian variables, and the piecewise components, $a_i$) are used to update the posterior model for each document (using a freshly drawn sample each step).

First and foremost, we note that the best baseline model (i.e., the *NVDM*) is more competitive when both the prior and posterior models are learnt together (i.e., the *G-NVDM*), as opposed to the fixed prior of Miao et al. (2015). However, we observe that integrating our proposed piecewise variables yields even better results in our document modeling experiments, substantially improving over the baselines. More importantly, in some cases, as in the 20 News-Groups and Reuters datasets, increasing the number of pieces from 3 to 5 can further reduce perplexity. Thus, we have achieved a new state-of-the-art perplexity on 20 News-Group task and — to the best of our knowledge – better perplexities on the CADE12 and RCV1 tasks compared to using a state-of-the-art model like the G-*NVDM*. Furthermore, we observe iterative inference yields yet a further boost in performance since the bound estimated is tighter, however, this form of inference is expensive and requires additional meta-parameters (e.g., a step-size, an early-stopping criterion, etc.). We remark a simpler, and more accurate, approach to inference would be to use importance sampling.

In Table 2, we examine the top ten highest ranked words given a query term, using the decoder parameter matrix (since the decoder is directly affected by the latent variables in our document models). It appears that the piecewise variables affect what is uncovered by the model with respect to the data, as each model returns different, but relevant results with respect to the query word. In our current examples, it appears that the *H-NVDM* with 5 pieces returns more general words. For example, in the case of "government", the baseline seems to value the plural form of the word (which is largely based on morphology) while the hybrid model actually pulls out meaningful terms such as "federal", "policy", and "administration". The case of "space" is interesting–the hybrid with 5 pieces seems to value two senses of the word–one related to "outer space" (e.g., "sun", "world", etc.) and another related to the dimensions of depth, height, and width within which things may exist and move (e.g., "area", "form", "scale", etc.).

## 6.2 DIALOGUE MODELING

We experiment with VHRED for dialogue modeling. This is a difficult problem, extensively studied in the recent literature (Ritter et al., 2011; Lowe et al., 2015; Sordoni et al., 2015; Li et al., 2016; Serban et al., 2016a). Related systems for dialogue response generation have recently gained a significant amount of attention from industry, with high-profile projects such as Google's SmartReply system (Kannan et al., 2016) and Microsoft's chatbot Xiaolice (Markoff & Mozur, 2015). Even more recently, Amazon has announced the Alexa Prize Challenge for the research community with the goal of developing a natural and engaging chatbot system (Farber, 2016).

We focus on non-goal-driven dialogue modeling and use the **Twitter Dialogue Corpus** (Ritter et al., 2011) based on public Twitter conversations. The dataset is split into training, validation, and test sets, containing respectively 749,060, 93,633 and 9,399 dialogues each. On average, each dialogue contains about 6 utterances (dialogue turns) and about 94 words. The dataset is the same as used by Serban et al. (2016b), but further pre-processed using byte-pair encoding (Sennrich et al., 2016) using a vocabulary consisting of 5000 sub-words.[9] The dialogues are substantially longer than recent large-scale language modeling corpora, such as the 1 Billion Word Language Model Benchmark (Chelba et al., 2014), which usually focus on modeling single sentences.

Parameter optimization was conducted with a learning rate of 0.0002 and mini-batches of size 40 or 80.[10] We use a variant of truncated back-propagation and apply gradient clipping (Pascanu et al., 2012). Model selection and early stopping — the only additional form of regularization employed for this set of experiments — are conducted using the validation lower-bound, estimated using one stochastic sample per validation example.

---

[9]In addition to applying byte-pair encoding, we filtered out 601 test dialogues so that no test dialogue context overlapped with the training or validation sets.

[10]We had to vary the mini-batch size to make the training fit on GPU architectures with low memory.

| Word Time-related | G-VHRED G-KL | H-VHRED G-KL | P-KL | Word Event-related | G-VHRED G-KL | H-VHRED G-KL | P-KL |
|---|---|---|---|---|---|---|---|
| monday | 3 | 5 | **10** | school | 9 | 16 | **50** |
| tuesday | 2 | 3 | **7** | class | 11 | 16 | **27** |
| wednesday | 4 | 11 | **13** | game | 20 | 26 | **41** |
| thursday | 2 | 3 | **9** | movie | 12 | 20 | **41** |
| friday | 9 | 18 | **26** | club | 13 | 22 | **28** |
| saturday | 6 | 6 | **13** | party | 8 | 10 | **32** |
| sunday | 2 | 2 | **9** | wedding | 7 | 13 | **23** |
| weekend | 8 | 16 | **32** | birthday | 12 | 20 | **23** |
| today | 18 | 28 | **56** | easter | 15 | 15 | **23** |
| night | 16 | 31 | **68** | concert | 7 | 16 | **20** |
| tonight | 32 | 36 | **47** | dance | 11 | 12 | **21** |

| Word Sentiment -related | G-VHRED G-KL | H-VHRED G-KL | P-KL | Word Acronyms, Punctuation Marks & Emoticons | G-VHRED G-KL | H-VHRED G-KL | P-KL |
|---|---|---|---|---|---|---|---|
| good | **72** | **73** | 44 | lol | **394** | **358** | 312 |
| love | **102** | **101** | 38 | omg | **52** | **45** | 19 |
| awesome | **26** | **44** | 39 | . | 386 | 558 | **1009** |
| cool | 14 | 28 | **29** | ! | **648** | **951** | 525 |
| haha | **132** | **101** | 75 | ? | **507** | **851** | 221 |
| hahaha | **60** | **48** | 24 | * | **108** | **54** | 19 |
| amazing | 14 | **38** | 33 | xd | **28** | **42** | 26 |
| thank | **137** | **153** | 29 | ♡ | **56** | **42** | 24 |

Table 3: Approximate posterior word encoding on Twitter. The numbers are computed by counting the number of times each word is among the 5 words with the largest sum of squared gradients of the Gaussian KL divergence (G-KL) and piecewise constant KL divergence (P-KL)

Similar to Serban et al. (2016b), we use a bidirectional GRU RNN *encoder*, where the forward and backward RNNs each have 1000 hidden units. We experiment with *context* RNN encoders with 500 and 1000 hidden units, and find that that 1000 hidden units reach better performance w.r.t. the variational lower-bound on the validation set. The *encoder* and *context* RNNs use layer normalization (Ba et al., 2016). We experiment with *decoder* RNNs with 1000, 2000 and 4000 hidden units (LSTM cells), and find that 2000 hidden units reach better performance. For the G-VHRED model, we experiment with latent multivariate Gaussian variables with 100 and 300 dimensions, and find that 100 dimensions reach better performance. For the H-VHRED model, we experiment with latent multivariate Gaussian and piecewise constant variables each with 100 and 300 dimensions, and find that 100 dimensions reach better performance. We follow the training procedure of Serban et al. (2016b): we drop words in the decoder with a fixed drop rate of 25% and multiply the KL terms in the variational lower-bound by a scalar, which starts at zero and linearly increases to 1 over the first 60,000 training batches.

We also experiment with an LSTM baseline model and a HRED baseline model (Serban et al., 2016a). For the LSTM model, we experiment with number of hidden units (LSTM cells) equal to 1000, 2000 and 4000 and find that 4000 hidden units perform best w.r.t. validation perplextiy. For the HRED model, we use the same *encoder* and *context* RNN architectures as the G-VHRED and H-VHRED models described earlier. We set the *encoder* RNN to have 1000 hidden units. We experiment with a *context* RNN with 500 and 1000 hidden units, and find that 1000 hidden units reach better performance. For the decoder RNN, we experiment with 1000 and 2000 hidden units (LSTM cells) and find that 2000 hidden units perform better.

**Approximate Posterior Analysis** Our hypothesis is that the piecewise constant latent variables are able to capture multi-modal aspects of the dialogue. Therefore, we evaluate the models by analyzing what information they have learned to represent in the latent variables. For each test dialogue with $n$ utterances, we condition each model on the first $n-1$ utterances and compute the latent posterior distributions using all $n$ utterances. We then compute the gradients of the KL terms of the multivariate Gaussian and piecewise constant latent variables w.r.t. each word in the dialogue. Since the words vectors are discrete, we compute the sum of the squared gradients w.r.t. each word embedding. The higher the sum of the squared gradients of a word is, the more influence it will have

on the posterior approximation (encoder model). For every test dialogue, we count the top $5$ words with highest squared gradients separately for the multivariate Gaussian and piecewise constant latent variables.[11]

The results are shown in Table 3. The piecewise constant latent variables clearly capture different aspects of the dialogue compared to the Gaussian latent variables. The piecewise constant variable approximate posterior encodes words related to time (e.g. weekdays and times of day) and events (e.g. parties, concerts, Easter). On the other hand, the Gaussian variable approximate posterior encodes words related to sentiment (e.g. laughter and appreciation) and acronyms, punctuation marks and emoticons (i.e. smilies). We also conduct a similar analysis on the document models evaluated in Sub-section 6.1, the results of which may be found in the Appendix.

**Response Evaluation** Non-goal-driven dialogue models are typically evaluated by asking humans to rate the quality of different responses. We follow the approach by Liu et al. (2016) by conducting an Amazon Mechanical Turk experiment to compare the G-VHRED and H-VHRED models. For each test dialogue, we use TF-IDF to extract 100 candidate responses (Lowe et al., 2015). We then rank the responses according to the G-VHRED model and H-VHRED model using the variational lower-bound.[12] We ask three human evaluators to rate model responses for $45$ dialogues on a Likert-type scale $1 - 5$, with $1$ representing an inappropriate response and $5$ representing a highly appropriate response.[13] For each dialogue, we show the human evaluators the top two responses ranked by the G-VHRED and H-VHRED models. We choose to evaluate the re-ranked responses for two reasons. First, it reduces variance in the output because it uses the approximate posterior model, compared to using beam search with samples from the high-entropy prior. Second, it decreases the number of generic responses, which are extremely common among generative models and which human evaluators tend to prefer despite not advancing the dialogue (Li et al., 2016).

The results are as follows. The G-VHRED model achieves scores $1.88$ and $2.13$ for the first and second ranked responses on average, and the H-VHRED model achieves scores $1.93$ and $2.04$ on average. In other words, H-VHRED performs nominally better on the first ranked response while G-VHRED performs nominally better on the second ranked response. In conclusion, if there exists a difference between the two models, naive human evaluators cannot see it.

Although naive human evaluators cannot distinguish between the model responses, based on our previous analysis we know that the two models encode different aspects of dialogue conversations. Therefore, we further investigate the probability of different responses to dialogue contexts related to time and events. Two examples are shown in Figure 2, where the dialogue contexts are *"when do you want to meet this weekend?"* and *"where are you going tomorrow?"*. H-VHRED assigns substantially more probability mass to relevant words compared to the G-VHRED as well as an LSTM baseline and HRED baseline. This confirms the ability of the piecewise constant latent variable to generate responses related to time and events.

Finally, we also evaluate the diversity of the G-VHRED and H-VHRED model outputs w.r.t. the top ranked FF-IDF candidate responses. We measure the average word entropy (Serban et al., 2016b) as well as number of unique words for each response and unique words across all test responses, but did not find a significant difference between the two models. This indicates that the Gaussian latent variables alone are able to increase response diversity, while the piecewise constant latent variables instead help encode specific aspects of the dialogue such as time and events.

---

[11]Our approach is equivalent to counting the top $5$ words with the highest L2 gradient norms. We also did some experiments using L1 gradient norms, which showed similar patterns.

[12]We use one stochastic sample.

[13]Human evaluators are only given a minimal description of the task, without any examples, before beginning the evaluation.

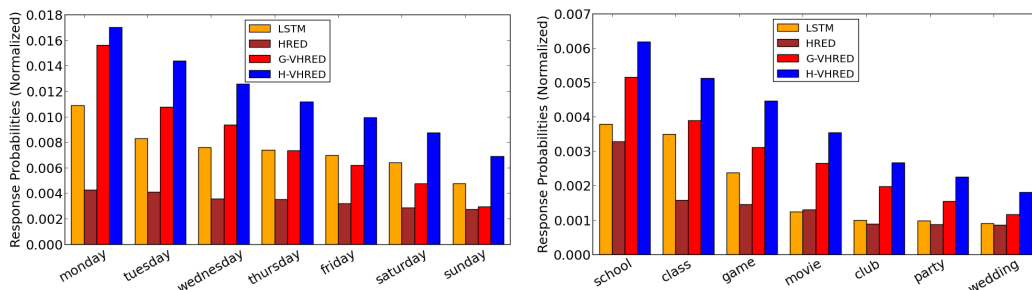

Figure 2: Probabilities for different responses related to time and events: left) probabilities for giving a one-word response with one of the weekdays (*monday, tuesday, . . . , sunday*) conditioned on the context utterance *"when do you want to meet this weekend?"*, right) probabilities forgiving a one-word response with one of several event-related nouns (*school, class, . . . , wedding*) on the context utterance *"where are you going tomorrow?"*. The probabilities have been normalized in log-space by the number of words in the response including end-of-utterance tokens. For G-VHRED and H-VHRED, the probabilities were estimated using the variational lower-bound over 10 samples.

## 7 CONCLUSIONS

In this paper, we have proposed the multi-modal variational encoder-decoder framework. In order to capture complex aspects of unknown data distributions, we developed the piecewise constant prior, which can be efficiently and flexibly adjusted to capture distributions with many modes, such as those over topics. In experiments on document modeling and dialogue modeling, we have shown the effectiveness of our framework in building models capable of learning richer structure from data. In particular, we have demonstrated new state-of-the-art results on several document modeling tasks.

Future work should focus on exploring other natural language processing tasks, where multi-modality plays an important role such as modeling technical help dialogues (Lowe et al., 2015) and online debates (Rosenthal & McKeown, 2015), and where additional information is available, such as in semi-supervised document categorization (Ororbia II et al., 2015a). Furthermore, the piecewise variables proposed in this work could prove useful in uncovering interesting and novel information in lesser-explored corpora.

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

## APPENDIX A: ANALYSIS OF DOCUMENT MODEL PIECEWISE VARIABLES

We present an additional analysis of the learned 20 News-Groups document models in order to explore what each set of latent variables might be capturing. To calculate the gradient of the KL terms needed to formulate word scores, we follow the approach described in Sub-section 6.2, however, conditioning only on the (training) document bag-of-words to compute the latent posterior to then calculate the gradient of the KL-terms with respect to each word in the document.

In Table 4, we observe results similar to those of Sub-section 6.2–the piecewise variables capture different aspects of the document data. It is worth noting, in this experiment, that the Gaussian variables alone were originally were sensitive to some of these words. However, in the hybrid model, nearly all of the temporal words that the Gaussian variables were once more sensitive to now more strongly affect the piecewise variables, which themselves also capture all of the words that were originally missed. This might indicate a shift in responsibility in which latent variables the document model decide are more suitable to capture certain aspects of the data. This effect appears to be even stronger in the case of certain nationality-based adjectives (e.g., "american", "israeli", etc.). While the G-NVDM could model multi-modality in the data to some degree, this work would be primarily done in the model's decoder. In the H-NVDM, the piecewise variables provide an explicit mechanism for capturing modes in the unknown target distribution, so it makes sense that the model would learn to use the piecewise variables instead, thus freeing up the Gaussian variables to capture other aspects of the data, as we found was the case with names (e.g., "jesus", "kent", etc.).

| Word Time-related | G-NVDM G-KL | H-NVDM-5 G-KL | P-KL | | Word Names | G-NVDM G-KL | H-NVDM-5 G-KL | P-KL |
|---|---|---|---|---|---|---|---|---|
| months | 23 | 33 | **40** | | henry | 33 | **47** | 39 |
| day | 28 | 32 | **35** | | tim | **32** | 27 | 11 |
| time | **55** | 22 | **40** | | mary | 26 | **51** | 30 |
| century | **28** | 13 | **19** | | james | 40 | **72** | 30 |
| past | **30** | 18 | **28** | | jesus | 28 | **87** | 39 |
| days | **37** | 14 | **19** | | george | 26 | **56** | 29 |
| ahead | **33** | 20 | **33** | | keith | 65 | **94** | 61 |
| years | **44** | 16 | **38** | | kent | 51 | **56** | 15 |
| today | 46 | 27 | **71** | | chris | 38 | **55** | 28 |
| back | 31 | 30 | **47** | | thomas | 19 | **35** | 19 |
| future | **20** | 15 | **20** | | hitler | 10 | **14** | 9 |
| order | 42 | 14 | **26** | | paul | 25 | **52** | 18 |
| minute | 15 | 34 | **40** | | mike | 38 | **76** | 40 |
| began | **16** | 5 | **13** | | bush | **21** | 20 | 14 |
| night | **49** | 12 | **18** | | | | | |
| hour | **18** | **17** | 16 | | **Adjectives** | **G-KL** | **G-KL** | **P-KL** |
| early | 42 | 42 | **69** | | american | **50** | 12 | **40** |
| yesterday | 25 | 26 | **36** | | german | **25** | 21 | **22** |
| year | **60** | 17 | **21** | | european | 20 | 17 | **27** |
| week | 28 | 54 | **58** | | muslim | 19 | 7 | **23** |
| hours | 20 | 26 | **31** | | french | 11 | **17** | **17** |
| minutes | **40** | 34 | **38** | | canadian | **18** | 10 | **16** |
| months | 23 | 33 | **40** | | japanese | 16 | 9 | **24** |
| history | **32** | 18 | **28** | | jewish | **56** | 37 | **54** |
| late | 41 | **45** | 31 | | english | 19 | 16 | **26** |
| moment | **23** | **17** | 16 | | islamic | 14 | 18 | **28** |
| season | **45** | 29 | 37 | | israeli | **24** | 14 | **18** |
| summer | 29 | 28 | **31** | | british | **35** | 15 | **17** |
| start | 30 | 14 | **38** | | russian | 14 | 19 | **20** |
| continue | 21 | 32 | **34** | | | | | |
| happened | 22 | 27 | **35** | | | | | |

Table 4: Approximate posterior word encodings on 20 News-Groups. For P-KL, we also bold every case where the piecewise variables showed greater sensitivity to the word than the Gaussian variables within the same hybrid model.

