# Peer review of "Multi-modal Variational Encoder-Decoders"

_ICLR 2017 — rejected_

[Public Comment · Christian A Naesseth · 07 Nov 2016]
**Differentiability of the reparameterization**

The reparameterization does not seem to be differentiable? I.e. eq. (8) consists of indicator functions of the parameters you are optimizing with respect to.

[Public Comment · (anonymous) · 07 Nov 2016]
**Is is ok to submit an incomplete paper?**

This paper is incomplete. Most of results are blank.  What is the meaning of "table XXXX"? 
Such strategy seems unfair....

However, I agree that methods are good.  This paper should be submitted to ICML or workshop in ICLR.

If this type of method is allowed, I would wonder the credibility of papers in this conference.

[Official Review · AnonReviewer3 · rating 4 · confidence 4 · 16 Dec 2016 (modified: 17 Dec 2016)]
**Official Review**

This paper proposes a piecewise constant parameterisation for neural variational models so that it could explore the multi-modality of the latent variables and develop more powerful neural models. 
The experiments of neural variational document models and variational hierarchical recurrent encoder-decoder models show that the introduction of the piecewise constant distribution helps achieve better perplexity on modelling documents and seemly better performance on modelling dialogues.
The idea of having a piecewise constant prior for latent variables is interesting, but the paper is not well-written (even 14 pages long) and the design of the experiments fails to demonstrate the most of the claims.  

The detailed comments are as follows:

--The author explains the limitations of the VAEs with standard Gaussian prior in the last paragraph of 3.1 and the last paragraph of 5.1. Hence, a multimodal prior would help the VAEs overcome the issues of optimisation. However, there is a lack of evidence showing the multimodality of the prior helps break the bottleneck. 

--In the last paragraph of 6.1, the author claimed the decoder parameter matrix is directly affected by the latent variables. But what the connects the decoder is a combination of a piecewise constant and Gaussian latent variables. No matter what is discovered in the experiments, it only shows z=

[Official Review · AnonReviewer1 · rating 3 · confidence 4 · 16 Dec 2016 (modified: 21 Dec 2016)]
**Review: Multi-modal Variational Encoder-Decoders**

UPDATE: I have read the authors' rebuttal and also the other comments in this paper's thread. My thoughts have not changed.

The authors propose using a mixture prior rather than a uni-modal
prior for variational auto-encoders. They argue that the simple
uni-modal prior "hinders the overall expressivity of the learned model
as it cannot possibly capture more complex aspects of the data
distribution."

I find the motivation of the paper suspicious because while the prior
may be uni-modal, the posterior distribution is certainly not.
Furthermore, a uni-modal distribution on the latent variable space can
certainly still lead to the capturing of complex, multi-modal data
distributions. (As the most trivial case, take the latent variable
space to be a uniform distribution; take the likelihood to be a
point mass given by applying the true data distribution's inverse CDF
to the uniform. Such a model can capture any distribution.)

In addition, multi-modality is arguably an overfocused concept in the
literature, where the (latent variable) space is hardly anymore worth
capturing from a mixture of simple distributions when it is often a
complex nonlinear space. It is unclear from the experiments how much
the influence of the prior's multimodality influences the posterior to
capture more complex phenomena, and whether this is any better than
considering a more complex (but still reparameterizable) distribution
on the latent space.

I recommend that this paper be rejected, and encourage the authors to
more extensively study the effect of different priors.

I'd also like to make two additional comments:

While there is no length restriction at ICLR, the 14 page document can
be significantly condensed without loss of describing their innovation
or clarity. I recommend the authors do so.

Finally, I think it's important to note the controversy in this paper.
It was submitted with many significant incomplete details (e.g., no experiments,
many missing citations, a figure placed inside that was pencilled in
by hand, and several missing paragraphs). These details were not
completed until roughly a week(?) later. I recommend the chairs discuss
this in light of what should be allowed next year.

[Official Review · AnonReviewer4 · rating 4 · confidence 5 · 20 Dec 2016]

The authors introduce some new prior and approximate posterior families for variational autoencoders, which are compatible with the reparameterization trick, as well as being capable of expressing multiple modes. They also introduce a gating mechanism between prior and posterior. They show improvements on bag of words document modeling, and dialogue response generation. The original abstract is overly strong in its assertion that a unimodal latent prior p(z) cannot fit a multimodal marginal int_z p(x|z)p(x)dz with a DNN response model p(x|z) ("it cannot possibly capture more complex aspects of the data distribution", "critical restriction", etc).

While the assertion that a unimodal latent prior is necessary to model multimodal observations is false, there are sensible motivations for the piecewise constant prior and posterior. For example, if we think of a VAE as a sort of regularized autoencoder where codes are constrained to "fill up" parts of the prior latent space, then there is a sphere-packing argument to be made that filling a Gaussian prior with Gaussian posteriors is a bad use of code space. Although the authors don't explore this much, a hypercube-based tiling of latent code space is a sensible idea.

As stated, I found the message of the paper to be quite sloppy with respect to the concept of "multi-modality." There are 3 types of multimodality at play here: multimodality in the observed marginal distribution p(x), which can be captured by any deep latent Gaussian model, multimodality in the prior p(z), which makes sense in some situations (e.g. a model of MNIST digits could have 10 prior modes corresponding to latent codes for each digit class), and multimodality in the posterior z for a given observation x_i, q(z_i|x_i). The final type of multimodality is harder to argue for, except in so far as it allows the expression of flexibly shaped distributions without highly separated modes. I believe flexible posterior approximations are important to enable fine-grained and efficient tiling of latent space, but I don't think these need to have multiple strong modes. I would be interested to see experiments demonstrating otherwise for real world data.

I think this paper should be more clear about the different types of multi-modality and which parts of their analysis demonstrate which ones. I also found it unsatisfactory that the piecewise variable analysis did not show different components of the multi-modal prior corresponding to different words, but rather just a separation between the Gaussian and the piecewise variables.

As I mention in my earlier questions, I found it surprising that the learned variance and mean for the Gaussian prior helps so dramatically with G-NVDM likelihood when the powerful networks transforming to and from latent space should make it scale-invariant. Explicitly separating out the contributions of a reimplemented base model, prior-posterior interpolation and the learned prior parameters would strengthen these experiments. Overall, the very strong improvements on the text modeling task over NVDM seem hard to understand, and I would like to see an ablation analysis of all the differences between that model and the proposed one.

The fact that adding more constant components helps for document modeling is interesting, and it would be nice to see more qualitative analysis of what the prior modes represent. I also would be surprised if posterior modes were highly separated, and if they were it would be interesting to explore if they corresponded to e.g. ambiguous word-senses.

The experiments on dialog modeling are mostly negative results, quantitatively. The observation that the the piecewise constant variables encode time-related words and the Gaussian variables encode sentiment is interesting, especially since it occurs in both sets of experiments. This is actually quite interesting, and I would be interested in seeing analysis of why this is the case. As above, I would like to see an analysis of the sorts of words that are encoded in the different prior modes and whether they correspond to e.g. groups of similar holidays or days.

In conclusion, I think the piecewise constant variational family is a good idea, although it is not well-motivated by the paper. The experimental results are very good for document modeling, but without ablation analysis against the baseline it is hard to see why they should be with such a small modification in G-NVDM. The fact that H-NVDM performs better is interesting, though. This paper should better motivate the need for different types of multi-modality, and demonstrate that those sorts of things are actually being captured by the model. As it is, the paper introduces an interesting variational family and shows that it performs better for some tasks, but the motivation and analysis is not clearly focused. To demonstrate that this is a broadly applicable family, it would also be good to do experiments on a more standard datasets like MNIST. Even without an absolute log-likelihood improvement, if the method yielded interpretable multiple modes this would be a valuable contribution.

[Final Decision · Program Chairs · 06 Feb 2017]
**ICLR committee final decision**

This paper explores a variational autoencoder variant.
 
 ICLR gives authors some respect that other conferences don't. It is flexible about the length of the paper, and allows revisions to be submitted. The understanding should be that authors should in turn treat reviewers with respect. The paper should still be finished. Reviewers can't be expected to read a churn of large revisions. The final paper should be roughly the right length, unless with very good reason.
 
 This paper was clearly not finished, and now is too long, with issues remaining. I hope that it will be submitted again, but not until it is actually ready.